# Cathepsin L Inhibitors with Activity against the Liver Fluke Identified From a Focus Library of Quinoxaline 1,4-di-*N*-Oxide Derivatives

**DOI:** 10.3390/molecules24132348

**Published:** 2019-06-26

**Authors:** Florencia Ferraro, Alicia Merlino, Jorge Gil, Hugo Cerecetto, Ileana Corvo, Mauricio Cabrera

**Affiliations:** 1Laboratorio de I + D de Moléculas Bioactivas, Departamento de Ciencias Biológicas, CENUR Litoral Norte, Universidad de la República, Paysandú 60000, Uruguay; 2Laboratorio de Química Teórica y Computacional, Instituto de Química Biológica, Facultad de Ciencias, Universidad de la República, Montevideo 11400, Uruguay; 3Laboratorio de Reproducción Animal, Producción y Reproducción de Rumiantes, Departamento de Ciencias Biológicas, CENUR Litoral Norte-Facultad de Veterinaria, Universidad de la República, Paysandú 60000, Uruguay; 4Grupo de Química Medicinal, Laboratorio de Química Orgánica & Área de Radiofarmacia, Centro de Investigaciones Nucleares, Facultad de Ciencias, Universidad de la República, Montevideo 11400, Uruguay

**Keywords:** *Fasciola hepatica*, cathepsin L, quinoxaline 1,4-di-*N*-oxides, small molecule inhibitors, molecular docking

## Abstract

Infections caused by *Fasciola* species are widely distributed in cattle and sheep causing significant economic losses, and are emerging as human zoonosis with increasing reports of human cases, especially in children in endemic areas. The current treatment is chemotherapeutic, triclabendazole being the drug of preference since it is active against all parasite stages. Due to the emergence of resistance in several countries, the discovery of new chemical entities with fasciolicidal activity is urgently needed. In our continuous search for new fasciolicide compounds, we identified and characterized six quinoxaline 1,4-di-*N*-oxide derivatives from our in-house library. We selected them from a screening of novel inhibitors against *Fh*CL1 and *Fh*CL3 proteases, two essential enzymes secreted by juvenile and adult flukes. We report compounds **C7**, **C17**, **C18**, **C19**, **C23,** and **C24** with an IC_50_ of less than 10 µM in at least one cathepsin. We studied their binding kinetics in vitro and their enzyme-ligand interactions in silico by molecular docking and molecular dynamic (MD) simulations. These compounds readily kill newly excysted juveniles in vitro and have low cytotoxicity in a Hep-G2 cell line and bovine spermatozoa. Our findings are valuable for the development of new chemotherapeutic approaches against fascioliasis, and other pathologies involving cysteine proteases.

## 1. Introduction

Liver flukes of *Fasciola* species infect cattle and sheep worldwide causing important economic global losses of over US$3 billion annually to the agricultural sector [1], and are responsible for increasing numbers of human infections, being recognized as an emerging human zoonosis by the World Health Organization. Alarmingly, reports of human cases in endemic areas of Asia, Africa, and South America appear frequently, especially in children, even considering that most human infections are thought to remain undiagnosed [2,3]. *Fasciola hepatica* has a complex life cycle and are adapted to hosts of multiple species [4], thus favoring the expansion of the parasite distribution and making it an infection that cannot be targeted for eradication. Since 1995, this is aggravated by several reports of parasite isolates resistant to triclabendazole, the drug of choice to treat humans and the only one that is effective against both the mature and juvenile forms of the parasite [5,6]. Also, despite sustained efforts, there is currently no available vaccine to prevent infection [7]. These facts highlight how important it is to design new control strategies to prevent and treat fascioliasis and find new targets for vaccines and drug development.

In this sense, cysteine proteases are key enzymes that play essential roles in the *F. hepatica* life cycle [8], showing functional specialization [9,10,11]. While cathepsin L3 and cathepsin B are highly expressed by newly excysted juveniles (NEJ) taking part in excystation and early parasite migration [12,13], a different set of enzymes are secreted by adult flukes (mainly cathepsin L1 and 2) to aid in feeding and immune modulation [14,15]. Cathepsins have been previously explored as interesting targets for antiparasitic chemotherapy in *Fasciola* [16,17], other helminth parasites [18,19], and several protozoa (like *Trypanosomes* and *Plasmodium falciparum*) [20,21,22,23]. Inhibitors targeting cysteine proteases have also been extensively studied as potential drugs for other diseases, since cathepsins are often found upregulated in cancer, osteoporosis, and atherosclerosis, among others. A large number of small molecule peptide-like cathepsin inhibitors have been synthesized, which include irreversible inhibitors such as epoxy succinic acid, vinyl sulfone, and acyl hydrazine derivatives and reversible ones, like cycloketones, aldehydes, and nitrile derivatives [24].

We recently identified and characterized synthetic flavonoids as non-peptide inhibitors of *Fasciola hepatica* cathepsin L and observed their trematocidal activity against juvenile parasites [16]. Non-peptide molecules are considered a better strategy for in vivo inhibition in order to avoid degradation by proteases. Continuing with our efforts to identify new active compounds against fascioliasis from our in-house chemical library we selected a series of twenty-eight quinoxaline 1,4-di-*N*-oxide derivatives in order to study their ability to inhibit essential cathepsin L enzymes from *Fasciola hepatica*. Most are synthetic, with only a few examples of natural ones, such as echinomycin and triostin-A. These kinds of molecules have been described as antitubercular, antimalarial, antileishmania, and antichagas, among other neglected diseases [25]. However, to the best of our knowledge, there is only one report of a quinoxaline derivative that inhibited *F. hepatica* thioredoxin glutathione reductase and killed NEJ in vitro [26]. The wide spectrum of activity could be attributed to the great versatility of the quinoxaline 1,4-di-*N*-oxide nucleus, which allows the generation of a large number of derivatives. The green chemistry methodologies available for their synthesis, which includes recyclable catalysts, microwave-assisted synthesis and water reactions, contribute to their attractiveness as drug development candidates [27]. Likewise, heterocyclic *N*-oxides have emerged as promising agents against several neglected and infectious diseases [28], and some quinoxaline derivatives have been reported as antihelminthic compounds against nematodes, cestodes, and trematodes [26,29].

In this work, we identified and characterized quinoxaline 1,4-di-*N*-oxide derivatives as novel inhibitors of the two main cathepsins secreted by juvenile and adult liver flukes. These compounds readily kill NEJ in vitro and have low cytotoxicity in a human cell line and bovine sperm. These findings open new avenues for the development of novel agents to control fluke infection and possibly other helminthic diseases.

## 2. Results and Discussion

### 2.1. Screening of Quinoxaline 1,4-di-N-Oxide Derivatives as FhCL1 and FhCL3 Inhibitors

We chose a set of twenty-eight quinoxaline 1,4-di-*N*-oxide derivatives with different substituents in R1–R4 (Table 1) that were tested as inhibitors of two *F. hepatica* cysteine proteases. In Table 1 we present the list of the assayed compounds and their chemical structures.

The percentage of inhibition of both enzymes for each compound is shown in Figure 1. We found eleven derivatives that inhibited at least one of the enzymes above 50%: **C6**, **C7**, **C11**, **C15**, **C17**, **C18**, **C19**, **C20**, **C21**, **,** and **C24**. Among these, three compounds stand out with an inhibition of 80% or more against both enzymes: **C17**, **C23,** and **C24**.

A shared feature of the compounds with the highest inhibition percentages is the presence of an electrophilic group at R1, like an ester, amide or nitrile, that may be susceptible to nucleophilic attack by the reactive thiol at the cathepsin active site. Also, these groups have a stereoelectronic analogy with peptide bonds, the cathepsin natural substrates. The only exception is **C11**, which has methyl groups in R1 and R2. However, its activity might be attributed to the presence of a *N*-phenethylsemicarbazone residue in R3. We observed that this substituent highly favors enzyme inhibition when comparing **C6** with **C7** and **C10** with **C11**, where it improves the inhibitory activity by more than 30% in both enzymes (Table 1 and Figure 1). Strikingly, all tested quinoxaline derivatives with a phenyl substituent in R2 (**C17**, **C18**, **C19**, **C23,** and **C24**) are among the best inhibitors. The positive contribution of the phenyl group in R2 to enzyme inhibition can be observed when comparing **C17** and **C15**, where the presence of a phenyl instead of a methyl group in R2 markedly increased inhibition of both enzymes. Moreover, the three derivatives with the highest cathepsin inhibition percentages (between 79–98%) have this substitution. Likewise, in our previous work, we found that chalcones substituted by a naphthyl group on both rings, were the best inhibitors of *F. hepatica* cathepsins [16], suggesting that the presence of bulky substituents and cyclic structures favor the inhibition of these enzymes.

Our results showed that the quinoxaline 1,4-di-*N*-oxide core contributes to the observed activity in a substitution-dependent manner. The principal pharmacophoric requirements that modulate cathepsin activity for this family of compounds are summarized in the scheme in Figure 2.

### 2.2. Screening of Cathepsin L Inhibitors against NEJ

We performed in vitro excystment of *F. hepatica* metacercariae to evaluate the effect of incubating the NEJ parasites with 50 μM of the cathepsin L inhibitors. We evaluated the parasite movement over a time-course period as an assessment of parasite vitality, together with microscopic examination of parasite morphology (Figure 3). The incubation with **C17**, **C18**, **C19**, **C21,** and **C24** resulted in a progressive decrease of parasite motility starting between 24 and 96 h of culture and increasing with time (Figure 3A) when clear signs of internal and tegument damage appeared (Figure 3B). After the NEJ stopped moving, a dark precipitate could be seen inside the gut, most notably in the presence of the quinoxaline **C17**, **C19,** and **C24** derivatives (Figure 3B).

Among the compounds that showed high inhibition of enzyme activity, we found that those derivatives with a phenyl substituent in R2 (**C17**, **C18**, **C19,** and **C24**) also possess strong fasciolicide activity in vitro, with the exception of **C23** that slightly affected parasite motility. The structure of these compounds share the presence of bulky substituents in R2 and R3/R4, while the good enzyme inhibitors substituted with methyl or amino groups in these positions (**C6**, **C7**, **C11,** and **C15**) did not have activity against the parasites (see Table 1 and Figure 3). The importance of bearing a phenyl group at the R2 position for the fasciolicide activity is clearly seen if we compare **C17** and **C15** derivatives. The IC_50_ for our best compound **C17** was estimated to be 25.1 μM (Figure 3C).

Our results with the in vitro cultured NEJ are consistent with our previous studies of cathepsin inhibitors, where we observed a reduction in NEJ motility and parasite death when incubated with chalcones that inhibit cathepsin Ls activity [16]. Also, reports of RNA interference against Cat-L and B and an in vitro treatment of NEJ with the cathepsin L and B inhibitors E64-d and CA-074Me, described a loss of worm motility accompanied by structural damage of the parasites [30,31], and a reduction in the ability to migrate through the duodenum wall [30]. In vivo, these results might translate into an inability of the parasites to migrate through the peritoneal cavity and into the liver, thus preventing the establishment of the infection.

### 2.3. Analysis of Hit Compounds

To further characterize these molecules, we selected six hit compounds: **C7**, **C17**, **C18**, **C19**, **C23,** and **C24** to determine their IC_50_ (Figure 4). We chose them based on the enzymatic and phenotypic screenings as they showed the highest ability to inhibit cathepsins in vitro and to impair NEJ development. We included **C7** and **C23** because although they did not have fasciolicidal activity, they strongly inhibited cathepsin activity.

In general, the compounds reached higher inhibition percentages with *Fh*CL1 than *Fh*CL3, and accordingly, the IC_50_ values were smaller for the former enzyme. All of the compounds had an IC_50_ of 10 μM or less with *Fh*CL1, while four of them had similar low values for *Fh*CL3 (**C7**, **C17**, **C18,** and **C24**). These results are consistent with our previous observations when screening chalcones for cathepsin L inhibition where we found that *Fh*CL1 activity was easier to disturb and was inhibited by a higher number of the compounds assayed when compared to *Fh*CL3 [16]. This could be a consequence of the broader conformation of *Fh*CL1 active site cleft that might accommodate different small molecules straightforwardly [33,34].

### 2.4. Cathepsin Inhibitors Are Slow-Binding and Interact at the Active Site Cleft

The calculated IC_50_ values give an idea of the binding affinity of the compounds for the parasite enzymes (Figure 4). A retrospective study of successful drugs suggests that kinetic parameters correlate better with in vivo efficacy than binding affinity [35]. We then studied whether the extent of compounds inhibition was time-dependent. We observed that compounds show a slow-binding interaction with the enzymes, some of them showing a different behavior with each enzyme. Compounds **C7**, **C17,** and **C23** required short times to inhibit *Fh*CL1 enzyme, while **C18** and **C23** reached high inhibition of *Fh*CL3 within a few minutes of incubation (Figure 5).

Drug-target interactions are influenced by the conformational dynamics of the protein and the ligand, and the complex formed when they bind. Therefore, to get a better picture of the enzyme-drug complex, we performed molecular docking and molecular dynamics simulations to predict the binding site and the predominant binding interactions of these quinoxalines with cathepsins. For *Fh*CL3, the most representative clusters of compounds were observed to interact deep inside the catalytic cleft (1st binding site), while in *Fh*CL1, only compound **C18** was positioned in a similar site. The other five compounds were located at a second site a little further from the S_2_ and S_3_ pockets but near the catalytic dyad occluding the cleft entrance (Figure 6). In any case, the position occupied by the compounds could hinder substrate entrance and accommodation along the active site, and thus, exert the observed inhibition of enzyme activity.

Among the hit compounds, **C17** showed high enzyme inhibition in vitro and the strongest fasciolicide activity against the cultured NEJ (Figure 1 and Figure 3). In order to obtain a more detailed view of the interactions that are established between this compound and the target proteins, we performed a molecular dynamics (MD) simulation study. To start the simulations, we employed the most populated cluster obtained for the compound in the molecular docking as an initial complex and analyzed the interaction between the enzymes and **C17** during 100 ns.

For *Fh*CL3, the ligand is positioned deep along the active site cleft and interacts with Cys 25 of the catalytic dyad and with the S_2_ and S_3_ pockets, which are involved in substrate accommodation and catalysis (Figure 7) [17,33,36]. Several hydrophobic interactions are seen between the thiosemicarbazone and Gly 66, Met 68, Ala 133, Val 157, and Ala 160 from the S_2_ and S_3_ pockets. Also, the compound establishes a hydrogen bond and a pi interaction with Trp 177, and an electrostatic interaction (pi-cation) with Trp 67 (Figure 7, right panels). In particular, this residue has proven to be crucial for enzyme activity [33]. In the MD simulation with *Fh*CL1, **C17** is placed closer to the catalytic dyad compared to the docking site. Its interaction with *Fh*CL1 occurs in a different orientation compared to *Fh*CL3, so that the ester and phenyl groups at R1 and R2 are near the catalytic dyad and not the thiosemicarbazone. Hydrophobic interactions are seen between the compound and Val 137, located near the *Fh*CL1 S_2_ pocket. Moreover, hydrophobic contacts and a hydrogen bond are established with residues that belong to the oxyanion hole, Gln 19, Trp 177, and Trp 181 (Figure 7, left panels), where the transition state of the reaction is stabilized [37]. The occlusion of the oxyanion hole would hinder the proper positioning of the substrate for catalysis. We observe that hydrophobic stacking interactions contribute significantly to the affinity of the compound for parasite cathepsins. Accordingly, the hit compounds share hydrophobic substituents in more than one position.

The disturbing of the protein structure upon substrate binding is also reflected by the changes in the frequency of intra-molecular protein hydrogen bonds that vary with the presence of **C17**. Many hydrogen bonds change their occupancy in the presence of the compound, for *Fh*CL3 this is particularly seen in the residues around the active site pockets (Appendix A). In *Fh*CL1, the configuration of the glycine-rich loop 1 (residues 52–67) that edges the S_3_ pocket is greatly modified, even though **C17** does not make direct contact with this region (Appendix A). Conformational changes around the catalytic cleft contribute to the interference with the enzyme’s activity [38].

### 2.5. Selectivity Assessment against Human Cathepsin L

Since parasitic cathepsins have structural homology with mammalian lysosomal cathepsins (the parasite hosts), we should not be surprised if the hit compounds affect to some extent the activity of the human enzyme as well. However, an interesting finding was that all of the tested compounds had a stronger inhibitory activity for the parasite enzymes, thus reducing the probability of unwanted off-target effects. For **C17**, the compound that most readily killed the NEJ, the selectivity index for the parasite enzymes over the human cathepsin was 9.3 and 2.6 for *Fh*CL1 and *Fh*CL3, respectively, demonstrating higher efficiency to inhibit the parasite enzymes (Figure 8).

If we compare the subsite configuration of the human and parasite enzymes, there are many residues that differ between them and particularly at the S_2_ and S_3_ pockets (Table 2), that might explain the differences we observed in the inhibition of the enzymes. Most of the residues involved in the interactions between *Fh*CL3 and **C17** vary between the enzymes (64, 67, 157, and 160). Also, residue 205 located at the bottom of the S_2_ pocket is Ala for *Hs*CL, while Val and Leu are found in *Fasciola* cathepsins, contributing to shape a wider site as compared to the parasite enzymes. The nature of residue 205 had proven to determine the specificity of the human cathepsin L and K for short peptide substrates and small molecule inhibitors [39]. As for *Fh*CL1, even though the oxyanion configuration is conserved between the three enzymes, the other residues interacting with *Fh*CL1 are variable in *Hs*CL. While *Fh*CL1 has Val at residue 137 and Met at residues 142 and 143, *Hs*CL has Gly, Leu, and Phe, respectively (Table 2).

### 2.6. Cytotoxicity Evaluation in Bovine Sperm and a Human Cell Line

We evaluated the cytotoxicity of the compounds in two cell models, assessing if they impair the viability of bovine spermatozoa and the human cell line HepG2. None of the compounds affected the sperm motility, and all of them have an IC_50_ for HepG2 between 4–50 µM (Table 3). For the HepG2 cells, the IC_50_ values for the active compounds are in the same order of magnitude of triclabendazole whose IC_50_ was 32 µM. This is the drug of reference to treating human infections and the only one that is effective in killing the juvenile parasites [5].

## 3. Conclusions

In this work, we performed a screening of twenty-eight quinoxaline 1,4-di-*N*-oxide derivatives with different substituents at R1–R4 to assess their ability to inhibit two *Fasciola hepatica* cathepsin Ls that take part in essential parasite processes. We found that those derivatives with a phenyl group at R2 and bulky substituents in R3/R4 potently inhibit both cathepsins and have strong fasciolicide activity over NEJ. The hit compounds interact in a slow-binding fashion with the enzymes in two different sites along the active site cleft through hydrogen bonds and hydrophobic interactions with functional enzyme residues at the oxyanion hole, catalytic dyad, and substrate binding subsites. The variability found at these sites among the human and parasitic cathepsin L might account for the higher inhibition attained against the *F. hepatica* enzymes. Interestingly, none of the compounds was toxic to bovine spermatozoa and the cytotoxicity level was similar to that of triclabendazole, the drug currently used for animal and human treatment. Our results will contribute towards the design of new drugs against cysteine proteases for the treatment of fascioliasis.

## 4. Materials and Methods

### 4.1. Production of Recombinant FhCL1 and FhCL3

*Fh*CL1 and *Fh*CL3 recombinant enzymes were expressed in the yeast *Hansenula polymorpha* as previously described [12,33]. Briefly, yeast transformants were cultured in 500 mL YEPD broth (1% glucose, 1% tryptone, 1% yeast extract) at 37 °C to an OD600 of 2–6, harvested by centrifugation at 3000× *g* for 10 min and induced by resuspending in 50 mL of buffered minimal media (0.67% yeast nitrogen base; 0.1 M phosphate buffer pH 6.0; 1% methanol) for 36 h at 30 °C. Recombinant pro-peptidases were secreted to the culture media and recovered by a 20–30-fold concentration of culture supernatants by ultrafiltration with a 10 kDa cut-off membrane. The proenzymes were autocatalytically activated to the mature form by incubation for 2 h at 37 °C in 0.1 M sodium citrate buffer (pH 5.0) with 2 mM dithiothreitol (DTT) and 2.5 mM EDTA, dialyzed against PBS pH 7.3, and stored in aliquots at −20 °C until used. The concentration of active enzyme was determined by titration against the cysteine protease inhibitor E-64c.

### 4.2. FhCL1 and FhCL3 Inhibition Screening

Each compound was completely dissolved in dimethylsulfoxide (DMSO) to a concentration of 10 mM to prepare the stock solutions. To evaluate the inhibition of *Fh*CL1 and *Fh*CL3 by quinoxaline 1,4-di-*N*-oxide derivatives, nanomolar concentrations of each enzyme was incubated with compounds at 10 μM. Briefly, each enzyme and compound were preincubated 20 min in a 96-well plate in 0.1 M sodium phosphate buffer pH 6, 1 mM DTT, and 1 mM EDTA at room temperature. The reaction was initiated by adding 20 μM of substrate and enzyme activity was measured by the increase in aminomethyl coumarin (AMC) fluorescence as peptide substrates were hydrolyzed (Z-VLK-AMC for *Fh*CL1 and Tos-GPR-AMC for *Fh*CL3), at an excitation wavelength of 340 nm and emission wavelength of 440 nm using a spectrofluorometer (Varioskan Thermo, Waltham, MA, USA). Enzyme activity was expressed as RFU/s (relative fluorescence units of AMC released per second). Each compound was tested in triplicate. A control reaction without enzymes was performed to check for the occurrence of non-catalyzed reactions between substrates and inhibitors and a spectrum from 300 to 450 nm wavelength was done for each quinoxaline 1,4-di-*N*-oxide derivative to verify that none of them had optical activity in the measurement range. The percentage of enzyme inhibition was calculated as 100 − (v_i_/v_o_) × 100, where v_i_ and v_o_ correspond to the initial rate of AMC fluorescence increase (RFU/s) with and without inhibitor, respectively.

### 4.3. In Vitro Culture of NEJ and Treatment with the Best Inhibitors

*F. hepatica* metacercariae were acquired from Instituto Miguel C. Rubino (DILAVE, MGAP, Uruguay). NEJ were obtained by in vitro excystement as previously described with minor modifications [40]. Briefly, metacercariae were incubated with 1% sodium hypochlorite for 5 min at room temperature to remove the outer cyst wall and then washed several times with PBS. Metacercariae activation was carried out in a medium prepared by mixing equal volumes of solution A (0.4% sodium taurocholate, 120 mM NaHCO_3_, 140 mM NaCl, pH 8.0) and solution B (50 mM HCl and 33 mM L-cysteine), and incubating at 39 °C until the NEJ begin to emerge (around 3–4 h). A 100 μm filter was used to retain the cyst wall. Collected NEJ were washed three times with RPMI-1640 supplemented with 200 U/mL Penicillin G sulfate, 200 mg/mL streptomycin sulfate, 500 ng/mL amphotericin B, 10 mM HEPES, counted and divided into groups of around 20 parasites that were transferred to 96 wells in tissue culture plates. Parasites were maintained at 37 °C, 5% CO_2_ in modified Basch’s medium [41]. At day 1, each tested compound was added at a concentration of 50 μM in 0.5% DMSO. Also, 0.5% of DMSO was added to the control groups. Each condition was tested in duplicate. NEJ behavior was monitored under a light microscope (Olympus BX41, Hamburg, Germany), every day each well was recorded for a minute in order to assess parasite motility and registered using the following score: 3—normally active; 2—reduced activity (sporadic movement); 1—immotile (dead, adapted from [32]). Control parasites were maintained in culture for seven days.

### 4.4. Characterization of the Best Inhibitors

#### 4.4.1. IC_50_, Time-Dependence of Inhibition, and HsCL Testing

To calculate the IC_50_ values, compounds were incubated with each enzyme at seven different concentrations; 0.625, 1.25, 2.5, 5, 10, 20, and 40 μM. Then, the enzyme activity was measured in a 96-well plate as described before. We plotted initial rates (RFU/s) versus log_10_ of inhibitor concentrations and obtained the IC_50_ values by fitting the data to a sigmoid curve. We also carried out slow-binding assays to evaluate the time-dependence of the inhibition. Each compound was incubated at 10 μM with the enzyme for increasing lengths of time; 3 to 120 min and the percentage of inhibition at each time point was determined as described before. We calculated the inhibition percentage of the leader compounds at 10 μM for *Hs*CL1. The enzyme was preincubated for 20 min with the compounds and the activity was measured in a 96-well plate in 400 mM sodium acetate, pH 5.5, with 4 mM EDTA, 8 mM DTT, and 20 μM of substrate Z-VLK-AMC. The percentage of enzyme inhibition was calculated as described for *Fh*CL1 and *Fh*CL3. We determined the IC_50_ value for **C17** against *Hs*CL by incubating the enzyme with four different concentrations of the compound 0, 10, 25, and 50 μM and measuring the activity as described before. We plotted initial rates (RFU/s) versus log_10_ of inhibitor concentration and obtained the IC_50_ value by fitting the data to a sigmoid curve.

#### 4.4.2. Ligand-Protein Molecular Docking to Predict the Binding Site into *Fh*CL1 and *Fh*CL3

##### Preparation of Protein Structures

*Fh*CL1 and *Fh*CL3 structures, previously obtained by homology modeling, were used [33,42]. In order to improve the accuracy of the structures for molecular docking, MD simulations were performed using the *pmemd* module implemented in the AMBER16 package [43], with the *ff14SB* force field [44]. Hydrogen atoms and sodium ions (to neutralize charge) were added to each protein with the *leap* utility. Each system was placed in a truncated octahedral box of TIP3P explicit water [45], and extended 10 Å outside the protein on all sides. The structures of *Fh*CL1 and *Fh*CL3 were treated as follows: (a) water and counterions were relaxed to minimize energy during 2500 steps (500 steepest descent steps, SD, and 2000 conjugate gradient steps, CG) with the protein restrained with a force constant of 500 kcal/molÅ 2; (b) the system was minimized without restraints during 20,000 steps (5000 SD and 15,000 CG). The long-range electrostatic interactions were considered using the particle-mesh Ewald (PME) method [46] and the non-bonded interactions cut-off of 10 Å was used. After minimization, each system was gradually heated in an NVT ensemble from 0 to 300 K over 100 ps using the Berendsen coupling algorithm [47]. This procedure was followed by 100 ns of NPT simulations at 300 K and 1 atm pressure using the Langevin dynamics algorithm [48]. All bonds involving hydrogen atoms were constrained using the SHAKE algorithm [49]. The equations of motion were integrated with a time step of 2.0 fs and coordinates of the systems were saved every 2 ps. Representative structures of *Fh*CL1 and *Fh*CL3 from the last 100 ns of the trajectories were obtained through cluster analysis using the average-linkage algorithm [50] and used for subsequent docking calculations. Clustering, root mean square deviation (RMSD), root mean square fluctuation (RMSF) and hydrogen bond analysis were performed using the cpptraj module in AmberTools16. For trajectory visualization, the Visual Molecular Dynamics (VMD) program was used [51].

##### Preparation of Ligand Structures

Compounds **C1**–**C28** (structures shown in Appendix A) were fully optimized at the ωB97XD/6-31G(d,p) level [52,53] in water using the integral equation formalism polarizable continuum model (IEF-PCM) [54] with the Bondi atomic radii and ultrafine grid. The nature of the optimized structures as stable species was inspected by checking the eigenvalues of the analytic Hessian matrix, calculated at the same level of theory, to be positive in all the cases. All these calculations were performed using the Gaussian09 software (Wallingford, CT, UK) [55].

##### Ligand-Protein Molecular Docking

Flexible-ligand docking was performed using a grid box of 126 × 126 × 126 points with a grid spacing of 0.45 Å in order to cover the entire protein surface (blind docking). The grid box was centered on the macromolecule. The results differing by less than 2.0 Å in the root-square deviation were grouped in the same cluster. The conformation with the lowest binding energy was chosen from the most populated cluster, and the corresponding ligand-protein complex was used for further MD studies. All docking calculations were done with the AutoDock 4.2 [56] software package using the Lamarckian genetic algorithm. A population size of 150 individuals and 2.5 × 10^6^ energy evaluations were used for 100 search runs.

#### 4.4.3. Ligand-Protein Molecular Dynamics MD Simulations

Ligand-protein molecular dynamics MD simulations of the five reversible compounds with the highest IC_50_ values were performed with FhCL1 and *Fh*CL3 as described above using the GAFF [57] force field for the ligand. RESP partial charges [58] for the compounds were derived using the Gaussian09 at the HF/6-31G level and the antechamber module in AMBER16 was employed to obtain the force field parameters. One-hundred nanoseconds of productive MD were simulated and coordinates of the systems were saved every 100 ps.

### 4.5. Cytotoxicity Assay on the HepG2 Cell Line and Bovine Spermatozoa

Cytotoxicity studies on the HepG2 cell line. Cells: 1.5 × 10^4^ cells per 96-well plate were cultured in 225 µL of DMEM medium, supplemented with L-glutamine (1%), penicillin/streptomycin (1%), and 10% (*v/v*) fetal bovine serum (FBS). The cultures were maintained at 37 °C and 5% CO_2_ for 48 h. Treatment: Compound solutions were prepared just before dosing. Stock solutions, 1 mM, were prepared in 100% DMSO (Aldrich, Saint Louis, MO, USA) and 25 µL of adequate dilution was added to each well 24 h after plating the cells. For each compound, a dose-response curve was done with five different compound concentrations: 6.25, 12.5, 25, 50, and 100 µM (treated cells, T). No effect on cell growth was observed by the presence of 0.5% DMSO in the culture media (control cells, C). Cells were incubated with the compounds for 24 h at 37 °C in 5% CO_2_ atmosphere, then the medium was discarded and the cells were washed with PBS. The cells were then fixed with 50 µL of TCA (50%) and 200 µL of culture medium (without FBS) for 1 h at 4 °C. Then the cells were washed with purified water and treated with Sulforhodamin B (0.4% wt/vol in 1% acetic acid) for 10 min at room temperature. The plates were then washed with 1% acetic acid and dried overnight. Finally, 100 µL of Tris buffer (pH = 10.0) was added and absorbance read at 540 nm. Data calculations: The assays were done in four replicates. At the end of the experiment, the cell survival percentage (SP) was calculated for each compound as (T/C) × 100. To determine the IC_50_, the SP of cells at each compound concentration was plotted and the data were fitted to a sigmoid curve. The standard error was not greater than 10% for any condition. For the cytotoxicity assay with bovine spermatozoa, semen samples were obtained from a healthy fertile Hereford bull and kept frozen in 0.5 mL straws (extended in Andromed, Minitube, Tiefenbach, Germany) under liquid nitrogen until use. The semen used belonged to a single freezing batch that was obtained during a regular collection schedule with an artificial vagina. Samples from three straws were thawed and a sperm pool was prepared in PBS at a concentration of 80 million spermatozoa per ml. Then, 50 μL of this pool was carefully mixed with 50 μL of each compound to be tested at a concentration of 50 µM or 1% DMSO in control experiments. Each compound was assayed in triplicate in 96-well plates and controls were assayed in triplicate. Plates were incubated at 37 °C for 1 h with moderate shaking. The motility analysis was carried out using a CASA (Computer Assisted Semen Analyzer) system Androvision (Minitube, Tiefenbach, Germany) with an Olympus BX 41 microscope (Olympus, Japan) equipped with a warm-stage at 37 °C. Each sample (10 μL) was placed onto a Makler Counting Chamber (depth 10 μm, Sefi-Medical Instruments, Haifa, Israel) and the following parameters were evaluated: percentage of total motile spermatozoa (motility >5 μm/s) and velocity curved line (VCL > 24 μm/s). At least 400 spermatozoa were analyzed from each sample from at least four microscope fields.

## Figures and Tables

**Figure 1 molecules-24-02348-f001:**
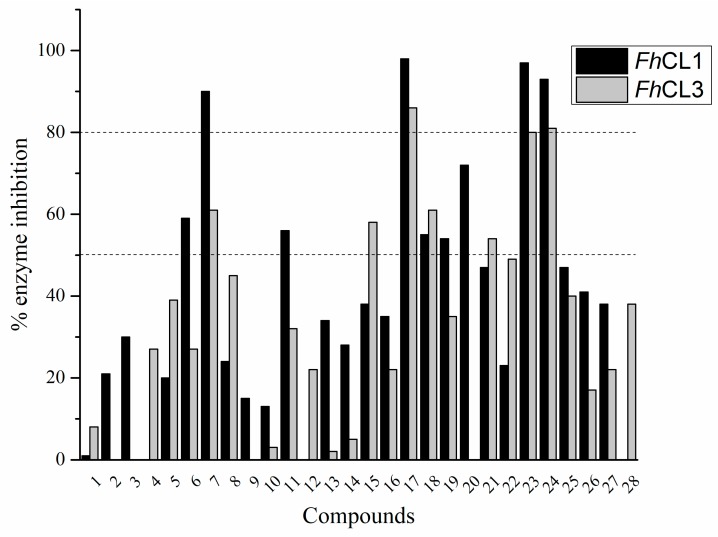
Screening of quinoxaline 1,4-di-*N*-oxide derivatives as *Fh*CL1 and *Fh*CL3 inhibitors. The percentage of inhibition of *Fh*CL1 is shown in black bars and that of *Fh*CL3 in gray bars. The standard deviation is less than 10% for all compounds.

**Figure 2 molecules-24-02348-f002:**
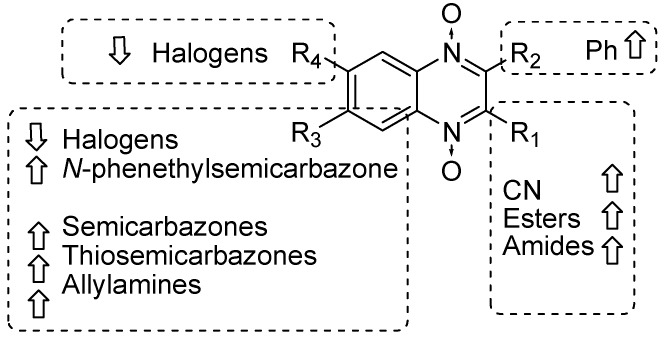
Schematic representation of the pharmacophoric requirements that modulate cathepsin enzyme activity. Ascending arrows indicate a substituent positively contributing to cathepsin inhibition while descending arrows indicate moieties that do not contribute to enzyme inhibition.

**Figure 3 molecules-24-02348-f003:**
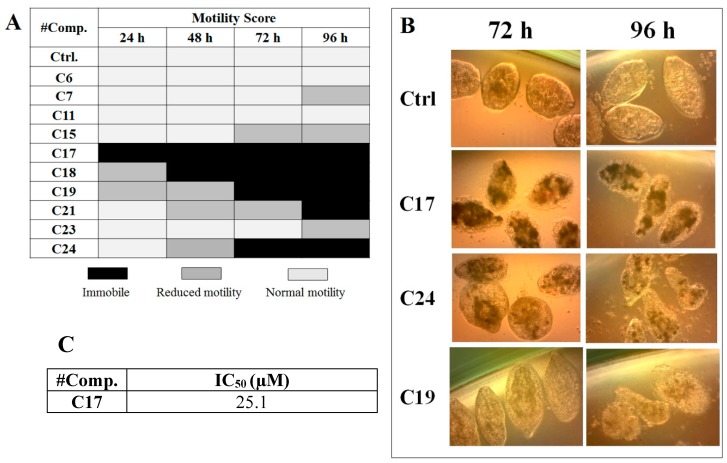
Effect of the cathepsin inhibitors against *F. hepatica* NEJ cultured in vitro. (**A**) Color chart showing the motility score of NEJ incubated with 50 μM of compound over 96 h. NEJ were classified in three-movement categories (taken from [32]: light gray = normal motility; gray = reduced motility (sporadic movement); black = immobile (dead). (**B**) Microscopic appearance of parasites incubated with 50 μM of **C17**, **C24,** and **C19** at 72 and 96 h treatment. Control NEJ were incubated in 0.5% DMSO. (**C**) IC_50_ determination for the most relevant compound, **C17**, after 48 h of incubation with NEJ.

**Figure 4 molecules-24-02348-f004:**
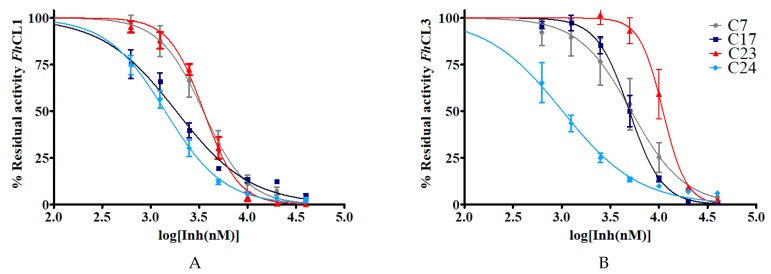
The IC_50_ of the hit quinoxaline 1,4-di-*N*-oxide derivatives are shown in μM. To obtain the IC_50_ value, the percentage of residual enzymatic activity for (**A**) *Fh*CL1 and (**B**) *Fh*CL3 was plotted against the logarithm of the concentration of compound (log[Inh(nM)]). The dose-response curves for the best inhibitors **C7**, **C17**, **C23,** and **C24** are shown below.

**Figure 5 molecules-24-02348-f005:**
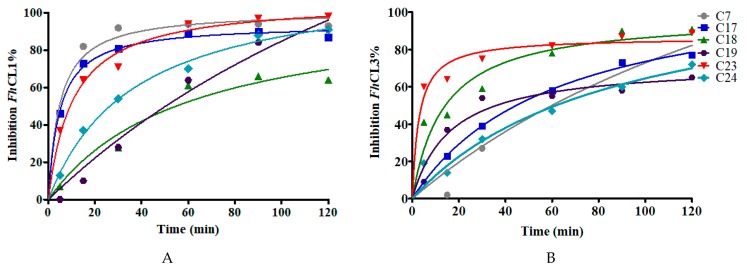
The effect of increasing the preincubation time of the compounds (at 10 μM) with the enzymes on the inhibition percentage to test for slow-binding kinetics. The enzymatic activity for (**A**) *Fh*CL1 and (**B**) *Fh*CL3 was measured after incubation of each enzyme with the inhibitors for different lengths of time from 5 to 120 min.

**Figure 6 molecules-24-02348-f006:**
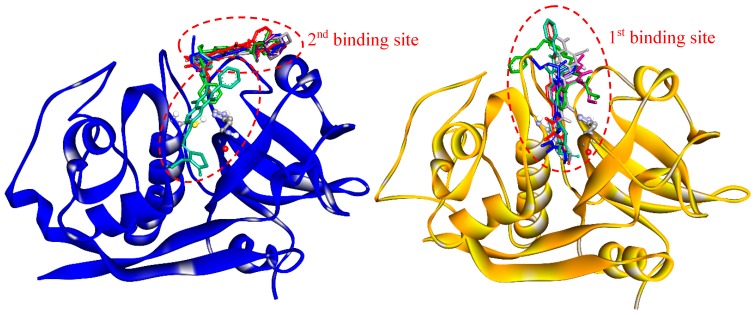
Binding sites of quinoxaline 1,4-di-*N*-oxide derivatives as predicted by molecular docking. Enzymes are represented in the new cartoon (*Fh*CL1 in blue and *Fh*CL3 in orange), the most representative clusters for the six-hit compounds are shown in sticks and the catalytic dyad in balls and sticks.

**Figure 7 molecules-24-02348-f007:**
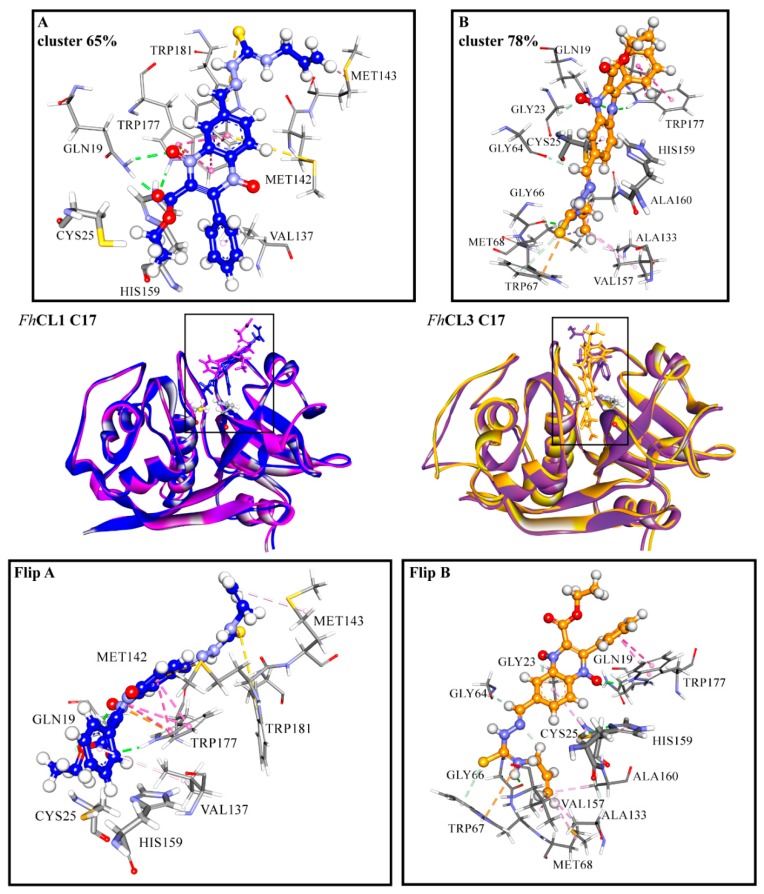
(**A**) *Fh*CL1 and (**B**) *Fh*CL3 interactions with **C17** from MD simulations. In the middle is shown the most representative cluster structure obtained in the MD simulations for each enzyme-ligand complex in blue for *Fh*CL1 and orange for *Fh*CL3, and the most representative cluster obtained by the docking of both enzymes in purple. We zoomed in the region of the enzyme-ligand interaction for the most representative cluster to show the enzyme (sticks) and ligand (scaled ball and sticks) residues involved and the interaction types in dashed lines (green: hydrogen bonds, pink: hydrophobic interactions, yellow: pi-sulfur, orange: pi-cation). In the lower panels the complex with **C17** for *Fh*CL1(Flip A) and *Fh*CL3 (Flip B) is seen from another perspective. The percentages correspond to the frequency obtained by the clustering of all the frames in each of the MD simulations. Residues are numbered according to papain (PDB code: 9PAP).

**Figure 8 molecules-24-02348-f008:**
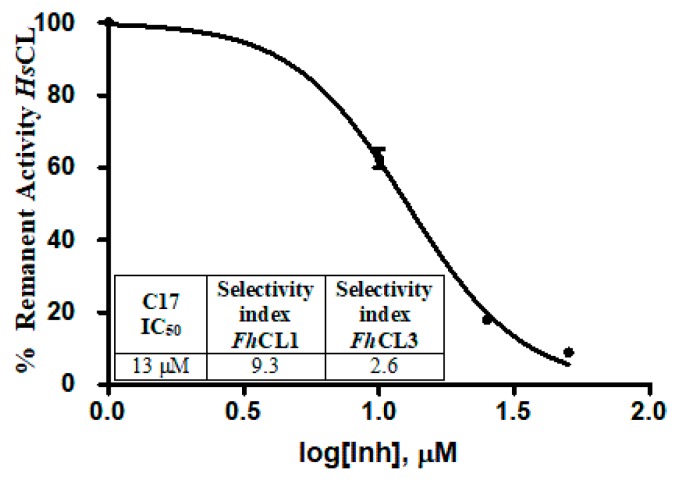
The table shows the comparison of the inhibition percentage of each compound at 10 μM for human cathepsin L (*Hs*CL) and the *Fasciola hepatica* enzymes. The standard deviation is less than 10% for each compound. In the graphic, a dose-response curve for **C17** was performed, and the IC_50_ and selectivity index for the parasite enzymes were calculated.

**Table 1 molecules-24-02348-t001:** Structures of quinoxaline 1,4-di-*N*-oxide derivatives evaluated as inhibitors of *Fasciola hepatica* cathepsins L1 and L3. Compounds were evaluated at 10 µM concentration, the percentage of inhibition is reported relative to the activity of the enzyme alone. The compounds with the highest inhibition are in bold. The whole structure of the compounds is given in Appendix A. The standard deviation is less than 10% for all compounds.

Quinoxalines General Structures	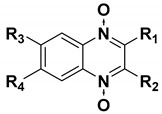
#Comp.	R1	R2	R3	R4	%Inh *Fh*CL1	%Inh *Fh*CL3
**C1**	-CH_3_	-H	-H	-H	1	8
**C2**	-CHO	-H	-H	-H	21	0
**C3**	-CN	-NH_2_	-H	-H	30	0
**C4**	-CN	-NH_2_	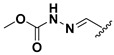	-H	0	27
**C5**	-CN	-NH_2_	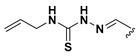	-H	20	39
**C6**	**-CN**	**-NH_2_**	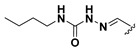	**-H**	**59**	**27**
**C7**	**-CN**	**-NH_2_**	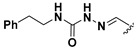	**-H**	**90**	**61**
**C8**	-CN	-NH_2_	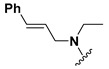	-H	24	45
**C9**	-CN	-NH_2_	-Cl	-Cl	15	0
**C10**	-CH_3_	-CH_3_	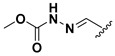	-H	13	3
**C11**	**-CH_3_**	**-CH_3_**	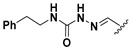	**-H**	**56**	**32**
**C12**	-CH_3_	-CH_3_	-H	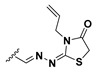	0	22
**C13**	-CH_3_	-CH_3_	-H	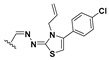	34	2
**C14**	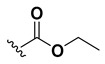	-CH_3_	-H	-H	28	5
**C15**	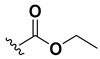	**-CH_3_**	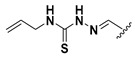	**-H**	**38**	**58**
**C16**	-CH_3_	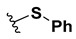	-Cl	-Cl	35	22
**C17**	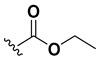	**-Ph**	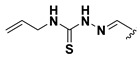	**-H**	**98**	**86**
**C18**	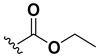	**-Ph**	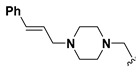	**-H**	**55**	**61**
**C19**	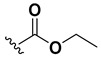	**-Ph**	**-Cl**	**-Cl**	**54**	**35**
**C20**	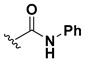	**-CH_3_**	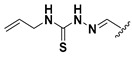	**-H**	**72**	**0**
**C21**	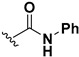	**-CH_3_**	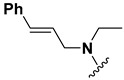	**-H**	**47**	**54**
**C22**	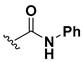	-CH_3_	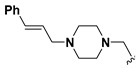	-H	23	49
**C23**	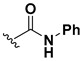	**-Ph**	**-H**	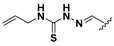	**97**	**79**
**C24**	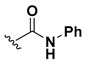	**-Ph**	**-H**	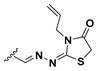	**93**	**81**
**C25**	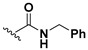	-CH_3_	-H	-H	47	40
**C26**	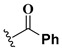	-CH_3_	-F	-F	41	17
**C27**	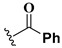	-CF_3_	-H	-H	38	22
**C28**	-CF_3_	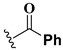	-F	-F	0	38

**Table 2 molecules-24-02348-t002:** Comparing the residues present in *Fh*CL1 (blue) and *Fh*CL3 (orange) that interact with compound **C17** between the *F. hepatica* enzymes and the *Hs*CL (PDB code: 1MHW). Residues are numbered according to papain (PDB code: 9PAP) as they appear in Figure 7, and the number corresponding to each enzyme is shown in superscripts. We indicated above if the residues belonged to the subsites.

Enzyme	S3			S2/S3				S2		
**Papain#**	**61**	**64**	**66**	**67**	**137**	**142**	**143**	**157**	**160**	**205**
***Fh*CL1**	**Asn^64^**	**Gly^67^**	**Gly^69^**	**Leu^70^**	**Val^139^**	**Met^144^**	**Met^145^**	**Val^161^**	**Ala^164^**	**Leu^210^**
***Fh*CL3**	**His^64^**	**Gly^67^**	**Gly^69^**	**Trp^70^**	**Ala^139^**	**Tyr^144^**	**Met^145^**	**Thr^161^**	**Ala^164^**	**Val^210^**
***Hs*CL**	**Glu^63^**	**Asn^66^**	**Gly^68^**	**Leu^69^**	**Gly^139^**	**Leu^144^**	**Phe^145^**	**Met^161^**	**Gly^164^**	**Ala^214^**

**Table 3 molecules-24-02348-t003:** Cytotoxicity assays in the human cell line HepG2 and bovine spermatozoa. For the HepG2 cell line, the IC_50_ was calculated for each compound. In bovine spermatozoa, compounds were tested at a fixed dose of 50 µM. Sperm motility was calculated relative to that of the control sperm (incubated with the vehicle). We included the drug of reference, triclabendazole, to test its cytotoxicity with both cell types.

Comp#	HepG2 IC_50_ (µM)	Sperm Motility RTC * (50 µM)
C17	48 ± 0.3	81 ± 3
C18	<6.2	82 ± 2
C23	12 ± 0.2	82 ± 1
C24	16 ± 0.1	82 ± 1
TCBZ	32 ± 0.2	88 ± 1

* RTC: relative to control.

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
