# Peer review of "Cathepsin L Inhibitors with Activity against the Liver Fluke Identified From a Focus Library of Quinoxaline 1,4-di-N-Oxide Derivatives"

_molecules, 2019, doi:10.3390/molecules24132348_

Round 1
Reviewer 1 Report
In my opinion the manuscript entitled: “Focus library of quinoxaline 1,4-di-N-oxide derivatives as cathepsin L inhibitors with activity against the liver fluke”, Authors: Florencia Ferraro, Alicia Merlino, Jorge Gil, Hugo Cerecetto, Ileana Corvo* and Mauricio Cabrera*, can be accepted after a minor revision.
In order to gain insights over developing a feasible chemotherapeutic approach for fighting fascioliasis, the work presents an interesting study, a combination of in silico methodologies (docking and molecular dynamics), along with in vitro study consisting in binding kinetics of quinoxaline 1,4-di-N-oxide derivatives to cathepsins. The experiments show that none of the compounds was toxic to bovine spermatozoa and the cytotoxicity level was within acceptable range.
The subject approached is of interest, the work has been well performed, the manuscript is well written and well-structured, and it will be significant for the scientific community.
I recommend publication. However, small editing corrections are required:
Please correct Figure 8, the last column of the table shows problems of editing.
Author Response
Please correct Figure 8, the last column of the table shows problems of editing
We corrected the table format in Figure 8.
Reviewer 2 Report
The authors report on the identification of several quinoxaline 1,4-di-N-oxide derivative molecules that act as inhibitors to two specific Fasciola cathepsin proteases. They analysed these molecules in vitro and modelled enzyme-ligand interactions in silico. These molecules were also shown to kill NEJ fluke. The work gives insights into the interactions between the inhibitors and protease active sites and may facilitate the development of chemotherapeutics against fascioliasis.
I’d change up the title, maybe something like: Identification of cathepsin L inhibitors with activity against liver fluke from a focus library of quinoxaline 1,4-di-N-oxide derivatives
Abstract is ok, but English (like in the rest of the manuscript) needs fixing.
Examples;
line 6: replace ‘.’ with ‘,’ after ‘Republica’
line 24: "..., triclabendazole being the drug of preference…..."
30: "essential" not "essentials"
33: "'the enzyme-ligand interactions..."
The content of the introduction is fine, providing adequate background on the parasite, the importance of parasite cathepsins, the inhibitor derivatives screened, and the work reported on.
However, the manuscript could do with another going over more carefully, or by someone with better English skills (native English speaker) to fix English errors throughout the manuscript. Typos & grammatical errors need to be fixed throughout the text, while some long (or poorly constructed) sentences should be split/rewritten to be clearer (eg. lines 69-72).
Some examples that I think need to be fixed (not exhaustive –other typos, grammatical errors, or poorly constructed sentences not listed), are listed below;
INTRO: 43: ...over US$3 billion annually...
43: I have problems with the reference here. Firstly the title of the reference doesn’t match your reference list – should it not be “A prospective view of animal and human Fasciolosis”. Seems this is mixed with next reference title on the reference list. Also your referencing something given in the summary of a review which provided no references itself. The review doesn’t provide any references for the statement of costs to agriculture sector, etc. Does it pluck this number from midair, or get it from somewhere? I assume it goes back to the costs calculated by Spithill et al 1997 (fasciolosis book chapter), but should reference it.
47-50: Fix this sentence - "Fasciola spp.". Also as spp. = species (plural), need to be specific about singular species (ie. “F. hepatica”) or fix the sentence. eg. has>have, is>are, etc.
54: change ‘imperious’ to more appropriate word eg. ‘important’
56: the F. hepatica…
65: ‘..and atherosclerosis…
66: ‘inhibitors, such as…’
69: ‘identified and characterized’. Also, this sentence needs fixing (lines 69-72).
75: Fix singular/plural issues in this sentence, eg. ‘molecule’ not ‘molecules’
77: Ross et al (2012) (doi: 10.1371/journal.pone.0035033) showed quinoxaline was inhibitor of F.hepatica TGR and killed fluke.
79:’..a large number…”
81-83: Fix this sentence. How sentence structured, it is not clear what “its” refers to – the quinoxaline 1,4-di-N-oxide nucleus, or derivatives.
Results & Discussion: The authors use a nice combination of in vitro and in silico studies to analyse protease inhibition with the tested compounds and the interactions between the inhibitors and proteases, presenting the results in a relatively clear manner. The conclusions drawn are supported by the data, but I felt that in sections the discussion & conclusion could have been written more clearly. I have a few minor comments regarding the discussion in this section, as stated below.
Further English fixes are needed in this section (typos, grammar, sentence rewrites). English fixes are also needed in methods. I have listed a few, but haven’t listed most English corrections needed.
110:’…against both enzymes’.
113: I’m not sure Figure 1 is necessary, as the inhibition data is already presented in Table 1.
116-118: Fix this sentence, eg- A shared feature of the compounds with the highest inhibition percentages was the presence of an electrophilic group at R1, like an ester, amide or nitrile, that may be susceptible to nucleophilic attack by the reactive thiol in the cathepsin active site.
116-: From your modelling, it doesn’t appear R1 is necessarily close to active site cysteine, so how important would susceptibility to nucleophilic attack be?. Also, is it really a shared feature, or just because of the 28 molecules you evaluated, most had this at R1, and of the R1 methyls, none had phenylsemicarbozone at R3 except C11 which was an ok inhibitor, and none had an R2 phenol, which appears very important for good inhibition.
119: ‘stereoelectronic’ misspelt.
129: You state above 80%. However, the 3 most inhibitory are all >93% vs CL1 but not >80% vs CL3. Please make statement clearer/more accurate.
134-146. While most inhibit CL1 more than CL3, I notice that the R3 allylamines tend to inhibit CL3 better than CL1 (C8, C18, C21, C22). I didn’t see any discussion of why this would be the case. Did modelling indicate anything relevant? Could the authors possibly elaborate on this?
Also C20 seems to be a very specific CL1 inhibitor, 72% vs 0% for CL3. The only difference between C20 and C21&22 (which are not CL1-specific inhibitors, and in fact inhibit CL3 more) is at R3. But I do not see any mention of this in the discussion. Also, I notice C20 is absent in NEJ assay. I would think that adding C20 to NEJ assay would be interesting – could almost act as a control to confirm decreased motility results from cathepsin L inhibition, as it shouldn’t inhibit CL3, so if any viability decrease seen, it suggests it results from inhibition of enzymes other than CL3.
138-140: Fix sentence.
157: I think ‘against’ better word than ‘over’ at several points in the manuscript. Eg. line 157, line 352.
158: ‘metacercariae’ misspelt (also line 400)
174: Are C15 and C17 a good example to use, as C15 has much lower inhibitory activity as stated in table 1, so wouldn’t this be expected to be less effective against NEJ in vitro? C23 had high inhibition, and R2 phenyl but only minimally effected NEJ viability. Also, as these cathepsins are typically secreted, would internalization be important for inhibition? Is internalization suggesting internal roles for cathepsins in parasite homeostasis?
Also, how were the inhibitors used in this assay chosen? If reduced NEJ viability is important and the cathepsins are temporally expressed with cathepsins at developmental timepoints having distinct functions (and enzyme active sites and specificities) wouldn’t those inhibitors that better act against CL3 (the juvenile specific cathepsin) be best to be evaluated eg, C8 and C22 are better inhibitors of CL3 than several of the inhibitors evaluated in the assay.
198: ‘…and determined their IC50…’
204: C18 has IC50 > 5uM. (=9.0)
212+216: Table needs to be labelled correctly.
216: Figure 4 table: Change commas to decimal points (‘,’ to ‘.’)
216: Fig 4. Maybe I am not reading this table correctly, but for FhCL1, how can the IC50 for C24 be higher than that of C17, when on the graph residual activity is consistently lower at equivalent concentrations?
221. ‘A retrospective…’
225-7: Is this statement true? From just looking at the figure it seems to me that they are fairly evenly split with C18,C19,C23 quicker for CL3, while C7,C17, C24 quicker with CL1.
239: Fig 6 & 7. Why, if inhibitor appears to sit deeper/better into FhCL3 active site cleft, are the compounds better inhibitors of FhCL1?
262+264:’thiosemicarbazone’ misspelt.
267: would not Smooker et al 2000 be an appropriate reference here, as first to demonstrate importance of this residue location in Fasciola cathepsin L proteases?
305: Figure 8: fix y axis title ‘remanent’
334 & 335, 500: ‘cytotoxicity’ misspelt
338-40: fix sentence
338: I think it is good you have included cytotoxicity data. But, as you stated in section 2.5, selectivity is surely important. Do you have any idea how TCBZ compares with your compounds in regards to effectiveness against NEJ? If they have similar IC50 values vs HepG2, this doesn’t mean much if TCBZ is used at doses 1000x less than this, but yours is needed at the same dose as this. Would this be future work you would look to do?
344: You state you included triclabendazole, but in the corresponding table 3, it is listed as ‘ND’ against sperm.
345: in methods (line 488) you state that the lowest concentration you tested was 6.25 uM. How can the IC50 for C18 be accurately stated at 4.4 uM if you didn’t test this low?
353 “…active site cleft…”
359-62: This sentence needs to be written more clearly, to better convey what you want to say.
Methods are thoroughly outlined and detailed. Some minor comments:
364: So you express the recombinant proteins into media, and you don’t need to purify it over a column of any sort?
406: change ‘in’ to ‘into’.
417-8: state 7 different concentrations, but actually list 8. Also log10 of the lowest concentration you tested, 1.25 uM is 3.10; so how is there a data point at ~2.8 on the figure 4 graphs? 0.625uM to 40uM perhaps?
417: “…depending on the solubility of the ...”
433: For 4.4.2-4.4.3 why not just refer to article [16] as much of this is word for word the same?
437-: ‘..the pmemd module..”; “…with the ff03.r1 force field…”; “the leap utility”” of FhCL1”
481: “1.5 x 104 cells
512: “depth 10 μm”
516: Add figure S2.
Figure legends: add spaces before ‘μM’ so matches manuscript text. Also do in other sections (Eg. section 4.2).
Supplementary material: “dashed squads”?; would “dashed boxes” be a better description?
Author Response
Thank you for your valuable suggestions, we included them in the manuscript and we believe is it greatly improved. We hope you can now consider it for publication.
Please find below the point by point reponse to your comments:
The authors report on the identification of several quinoxaline 1,4-di-N-oxide derivative molecules that act as inhibitors to two specific Fasciola cathepsin proteases. They analysed these molecules in vitro and modelled enzyme-ligand interactions in silico. These molecules were also shown to kill NEJ fluke. The work gives insights into the interactions between the inhibitors and protease active sites and may facilitate the development of chemotherapeutics against fascioliasis.
I’d change up the title, maybe something like: Identification of cathepsin L inhibitors with activity against liver fluke from a focus library of quinoxaline 1,4-di-N-oxide derivatives
We modified the manuscript title as suggested
Abstract is ok, but English (like in the rest of the manuscript) needs fixing.
Examples;
line 6: replace ‘.’ with ‘,’ after ‘Republica’
line 24: "..., triclabendazole being the drug of preference…..."
30: "essential" not "essentials"
33: "'the enzyme-ligand interactions..."
The content of the introduction is fine, providing adequate background on the parasite, the importance of parasite cathepsins, the inhibitor derivatives screened, and the work reported on.
However, the manuscript could do with another going over more carefully, or by someone with better English skills (native English speaker) to fix English errors throughout the manuscript. Typos & grammatical errors need to be fixed throughout the text, while some long (or poorly constructed) sentences should be split/rewritten to be clearer (eg. lines 69-72).
Some examples that I think need to be fixed (not exhaustive –other typos, grammatical errors, or poorly constructed sentences not listed), are listed below;
We conducted a thorough review of English spelling and writing. We corrected all the typing errors we found and re-wrote the sentences that were too long. We highlighted all the changes in yellow.
INTRO: 43: ...over US$3 billion annually...
43: I have problems with the reference here. Firstly the title of the reference doesn’t match your reference list – should it not be “A prospective view of animal and human Fasciolosis”. Seems this is mixed with next reference title on the reference list. Also your referencing something given in the summary of a review which provided no references itself. The review doesn’t provide any references for the statement of costs to agriculture sector, etc. Does it pluck this number from midair, or get it from somewhere? I assume it goes back to the costs calculated by Spithill et al 1997 (fasciolosis book chapter), but should reference it.
Yes, there was a mistake here. The title of the first reference was mixed with the second one as you said. We changed it for the suggested reference.
47-50: Fix this sentence - "Fasciola spp.". Also as spp. = species (plural), need to be specific about singular species (ie. “F. hepatica”) or fix the sentence. eg. has>have, is>are, etc.
54: change ‘imperious’ to more appropriate word eg. ‘important’
56: the F. hepatica…
65: ‘..and atherosclerosis…
66: ‘inhibitors, such as…’
69: ‘identified and characterized’. Also, this sentence needs fixing (lines 69-72).
75: Fix singular/plural issues in this sentence, eg. ‘molecule’ not ‘molecules’
77: Ross et al (2012) (doi: 10.1371/journal.pone.0035033) showed quinoxaline was inhibitor of F.hepatica TGR and killed fluke.
In this work three quinoxaline derivatives were tested and the only active one was not a 1,4 di-N-oxide, but a reduced derivative. However, we modified the sentence in the manuscript to be more accurate about what is reported in this article.
79:’..a large number…”
81-83: Fix this sentence. How sentence structured, it is not clear what “its” refers to – the quinoxaline 1,4-di-N-oxide nucleus, or derivatives.
We corrected the sentence, “its” referred to the quinoxaline derivatives.
Results & Discussion: The authors use a nice combination of in vitro and in silico studies to analyse protease inhibition with the tested compounds and the interactions between the inhibitors and proteases, presenting the results in a relatively clear manner. The conclusions drawn are supported by the data, but I felt that in sections the discussion & conclusion could have been written more clearly. I have a few minor comments regarding the discussion in this section, as stated below.
Further English fixes are needed in this section (typos, grammar, sentence rewrites). English fixes are also needed in methods. I have listed a few, but haven’t listed most English corrections needed.
110:’…against both enzymes’.
113: I’m not sure Figure 1 is necessary, as the inhibition data is already presented in Table 1.
Although the data is twice, we think that the table allows to better compare the inhibition with the structure of the compounds while the figure allows to see the global results, like there are more compounds inhibiting FhCL1 than FhCL3, more difficult to appreciate in the table. For this reason we keep the figure 1.
116-118: Fix this sentence, eg- A shared feature of the compounds with the highest inhibition percentages was the presence of an electrophilic group at R1, like an ester, amide or nitrile, that may be susceptible to nucleophilic attack by the reactive thiol in the cathepsin active site.
116-: From your modelling, it doesn’t appear R1 is necessarily close to active site cysteine, so how important would susceptibility to nucleophilic attack be?. Also, is it really a shared feature, or just because of the 28 molecules you evaluated, most had this at R1, and of the R1 methyls, none had phenylsemicarbozone at R3 except C11 which was an ok inhibitor, and none had an R2 phenol, which appears very important for good inhibition.
We studied the distances of the electrophilic centre of R1 and Cys25-SH in the docking. For FhCL1 that distance for esters and nitriles is between 8 and 15 Å. However amides are nearer, about 3-5 Å, a distance at which reaction might occur. None is near Cys25-SH in FhCL3, as compounds are in an inverse orientation, so R1 and R2 are away from the catalytic dyad. But the electrophilic centre form the thiosemicarbazone in R3 is between 3-5 Å. Not only distances are important for reactivity but also orientation and we can not see that in the docking studies. We keep the sentence “that may be susceptible to nucleophilic attack by the reactive thiol in the cathepsin active site”, just as a hypothesis.
119: ‘stereoelectronic’ misspelt.
129: You state above 80%. However, the 3 most inhibitory are all >93% vs CL1 but not >80% vs CL3. Please make statement clearer/more accurate.
We made this sentence more accurate
134-146. While most inhibit CL1 more than CL3, I notice that the R3 allylamines tend to inhibit CL3 better than CL1 (C8, C18, C21, C22). I didn’t see any discussion of why this would be the case. Did modelling indicate anything relevant? Could the authors possibly elaborate on this?
It is a nice observation, of those compounds we only did MD for C18 and C21 (not included in the manuscript). For FhCL3 we found an increment in enzyme flexibility (comparing RMSF between enzyme alone and complexed with compounds) not notorious for FhCL1, which may reflect more drastic changes in the structure of FhCL3 upon compound binding. However we don´t have an explanation of why this might be happening.
Also C20 seems to be a very specific CL1 inhibitor, 72% vs 0% for CL3. The only difference between C20 and C21&22 (which are not CL1-specific inhibitors, and in fact inhibit CL3 more) is at R3. But I do not see any mention of this in the discussion. Also, I notice C20 is absent in NEJ assay. I would think that adding C20 to NEJ assay would be interesting – could almost act as a control to confirm decreased motility results from cathepsin L inhibition, as it shouldn’t inhibit CL3, so if any viability decrease seen, it suggests it results from inhibition of enzymes other than CL3.
We didn´t include C20 because we looked for inhibitors of both enzymes. Just inhibiting FhCL3 the juvenile enzyme would not be so pharmacologically relevant as it is expressed over a very short period of time by NEJ. It is interesting to inhibit both in order to avoid NEJ reaching the liver and kill adults of a current infection with the same drug.
138-140: Fix sentence.
157: I think ‘against’ better word than ‘over’ at several points in the manuscript. Eg. line 157, line 352.
158: ‘metacercariae’ misspelt (also line 400)
174: Are C15 and C17 a good example to use, as C15 has much lower inhibitory activity as stated in table 1, so wouldn’t this be expected to be less effective against NEJ in vitro? C23 had high inhibition, and R2 phenyl but only minimally effected NEJ viability. Also, as these cathepsins are typically secreted, would internalization be important for inhibition? Is internalization suggesting internal roles for cathepsins in parasite homeostasis?
We took the example of C15 and C17 because they only differ at R2 and other moderate inhibitors like C15 (but with phenyl at R2 like C18 or C19) did kill the parasite, so besides inhibiting the enzymes, the phenyl at R2 seems important for the biological activity. Yes, C23 really surprised us. But we don´t know why is not killing the parasites as the other phenyl R2 compounds do. Indeed, we believe that compounds affect cathepsins before being secreted, interfering with their intracellular functions too. Also, as these parasites have analogous enzymes uncharacterized (like CL4 and CL6 in NEJ) there could be cross reactivity.
Also, how were the inhibitors used in this assay chosen? If reduced NEJ viability is important and the cathepsins are temporally expressed with cathepsins at developmental timepoints having distinct functions (and enzyme active sites and specificities) wouldn’t those inhibitors that better act against CL3 (the juvenile specific cathepsin) be best to be evaluated eg, C8 and C22 are better inhibitors of CL3 than several of the inhibitors evaluated in the assay.
Well, our idea was to select good inhibitors of both enzymes, thinking that they could be active against both stages of the parasite, NEJ and adults. We would like to evaluate these compounds in adult parasites to see if they have biological activity too.
198: ‘…and determined their IC50…’
204: C18 has IC50 > 5uM. (=9.0)
We modified the sentence to more accurately describe the results
212+216: Table needs to be labelled correctly.
216: Figure 4 table: Change commas to decimal points (‘,’ to ‘.’)
216: Fig 4. Maybe I am not reading this table correctly, but for FhCL1, how can the IC50 for C24 be higher than that of C17, when on the graph residual activity is consistently lower at equivalent concentrations?
Sorry about this, the values were exchanged by mistake, the IC50 for C17 is 1.7 and for C24 it is 1.4. We corrected them in the table
221. ‘A retrospective…’
225-7: Is this statement true? From just looking at the figure it seems to me that they are fairly evenly split with C18,C19,C23 quicker for CL3, while C7,C17, C24 quicker with CL1.
We modified the manuscript to better describe what is seen
239: Fig 6 & 7. Why, if inhibitor appears to sit deeper/better into FhCL3 active site cleft, are the compounds better inhibitors of FhCL1?
We re-wrote part of the results and discussion section regarding the interactions of C17 with enzymes. In FhCL1 the compound is far from S2 and S3 pockets compared to FhCL3, but is near the catalytic dyad. In the MD simulation it can be seen nearer Cys25 and His159. This was not clear in the manuscript, giving the impression that FhCL1 was completely away from the active site. In addition, we improved Figure 8 by adding one more panel for each enzyme where interactions are viewed from another perspective to aid visualization. There were some weak hydrogen bonds (such as C-H --- O) that were removed not to be confused with the strong ones (such as N-H --- O). We believe that now the interactions are better seen and that FhCL1 is strongly inhibited because although it does not directly interact with S2 and S3 subsite residues, it does interact with several residues of the enzyme in the vicinity of the catalytic dyad and the oxyanion hole.
262+264:’thiosemicarbazone’ misspelt.
267: would not Smooker et al 2000 be an appropriate reference here, as first to demonstrate importance of this residue location in Fasciola cathepsin L proteases?
We included this reference.
305: Figure 8: fix y axis title ‘remanent’
334 & 335, 500: ‘cytotoxicity’ misspelt
338-40: fix sentence
338: I think it is good you have included cytotoxicity data. But, as you stated in section 2.5, selectivity is surely important. Do you have any idea how TCBZ compares with your compounds in regards to effectiveness against NEJ? If they have similar IC50 values vs HepG2, this doesn’t mean much if TCBZ is used at doses 1000x less than this, but yours is needed at the same dose as this. Would this be future work you would look to do?
TCBZ is usually used at 10 mg/kg. TCBZ MW is 359.6 g/mol. We roughly estimated the concentration used to be around 30 µM.
344: You state you included triclabendazole, but in the corresponding table 3, it is listed as ‘ND’ against sperm.
We repeated the cytotoxicity evaluation in sperm including TCBZ, we included the compounds again to test them together with TCBZ in the same batch of sperm. The %motility are a little lower than the first experiment, but we still observed very low sperm toxicity for our compounds and TCBZ as well.
345: in methods (line 488) you state that the lowest concentration you tested was 6.25 uM. How can the IC50 for C18 be accurately stated at 4.4 uM if you didn’t test this low?
The IC50 value was extrapolated by the Prism software after fitting the data points to a sigmoid curve. But is true that the value is not accurate, we changed it for <6.25.
353 “…active site cleft…”
359-62: This sentence needs to be written more clearly, to better convey what you want to say.
We reformulated the sentence
Methods are thoroughly outlined and detailed. Some minor comments:
364: So you express the recombinant proteins into media, and you don’t need to purify it over a column of any sort?
We do not need to column purify it because we only see the bands corresponding to the proenzyme and the mature form of the enzymes in a SDS-PAGE gel.
406: change ‘in’ to ‘into’.
417-8: state 7 different concentrations, but actually list 8. Also log10 of the lowest concentration you tested, 1.25 uM is 3.10; so how is there a data point at ~2.8 on the figure 4 graphs? 0.625uM to 40uM perhaps?
We included 0, which is not a concentration that´s why we said 7 and list 8 numbers. Is confusing so we removed it. Yes, we tested from 0.625 uM to 40 uM, we corrected this in M&M section.
417: “…depending on the solubility of the ...”
433: For 4.4.2-4.4.3 why not just refer to article [16] as much of this is word for word the same?
Because there were some minor modifications
437-: ‘..the pmemd module..”; “…with the ff03.r1 force field…”; “the leap utility”” of FhCL1”
481: “1.5 x 104 cells
512: “depth 10 μm”
516: Add figure S2.
Figure legends: add spaces before ‘μM’ so matches manuscript text. Also do in other sections (Eg. section 4.2).
Supplementary material: “dashed squads”?; would “dashed boxes” be a better description?
We corrected all the suggested typo errors and went through the whole manuscript again correcting them. We also split in two the very long sentences.
Reviewer 3 Report
The authors should have been more careful making the changes since they have made several mistakes They must correct them in order to be published the manuscript.
In figure 7, the ligand in orange is shown in ball and sticks. The size of the balls should be smaller to clearer the figure.
They have to pay attention to the amino-acids labels. For FhCL1 C17 amino-acids Trp185 and Trp189 were Trp177 and Trp181 in the previous version of the manuscript. Besides, they mention Trp177 in the text but it is not in the figure 7. They also mention in the text His159 and they use HIE159 in the figure. This is also repeated in the figure S2.
In line 280 Figure S1 should be Figure S2. Where is figure 1S in the text?
The number of Figure 2S is missed in the supplementary information.
Author Response
Thank you for your useful suggestions. We think they have greatly improved the manuscript.
Here´s the point by point answers to your suggestions:
The authors should have been more careful making the changes since they have made several mistakes They must correct them in order to be published the manuscript.
We corrected them, changes in the manuscript are highlighted in yellow
In figure 7, the ligand in orange is shown in ball and sticks. The size of the balls should be smaller to clearer the figure.
To reduce their size we changed the “scaled ball and stick” format to “ball and stick”, we also included another panel of each enzyme to better show the interactions.
They have to pay attention to the amino-acids labels. For FhCL1 C17 amino-acids Trp185 and Trp189 were Trp177 and Trp181 in the previous version of the manuscript. Besides, they mention Trp177 in the text but it is not in the figure 7. They also mention in the text His159 and they use HIE159 in the figure. This is also repeated in the figure S2.
We corrected the Figures
In line 280 Figure S1 should be Figure S2. Where is figure 1S in the text?
We corrected this
The number of Figure 2S is missed in the supplementary information.
We included it
Reviewer 4 Report
there are some flaws in the manuscript that have been highlighted in the previous revisions that would require attention by the authors and that have not been persistently taken into consideration.
Author Response
Thanks for your comments, here are our point by point responses:
there are some flaws in the manuscript that have been highlighted in the previous revisions that would require attention by the authors and that have not been persistently taken into consideration.
Indeed, the authors report “It is very difficult and expensive to obtain
metacercarie to perform the experiments with the NEJ, that's why we
evaluated the compounds with the parasites at a fixed dose. There are
very few metacercarie suppliers in the world and currently, our supplier
in Uruguay, is not able to provide us more metacercarie.” This is a very
impressive response. I believe that a trained scientist should be able
to schedule and program the experiments and to acquire the necessary
materials in order to complete that certain experiment. In absence of
more in depth experiments over parasites for at least one optimal
compound, the information collected so far in this manuscript are too
preliminary and publication is not recommended.
We included the IC50 value for C17 in Figure 3 in our previous re-submission.
Second reply: “We agree with the comments but also believe that some
roughly observations can be made. Following your recommendation, we
erased the second paragraph of the SAR discussion. We kept the first one
because we consider that even if we used a single concentration, the
observed tendencies are based on the activity of several compounds”. It
is not possible to extrapolate such a kind of SAR comments as those
reported in the manuscript by using a single concentration. This part of
the manuscript must be removed or significantly reduced.
We re-wrote this paragraph and included the suggestions of the other reviewers as well.
“We used the log of the concentration of the compounds to fit the data
to a sigmoid curve to calculate the IC50”. So it has been used the
nanoMolar log concentration? This is quite peculiar, usually log
concentration are reported in Molar concentration.
If we use Molar we get negative numbers when calculating the logarithm, that´s why we used nanomolar. We found it easier and visually more friendly to see in the graphic.
Page 7 line 171: to my opinion what the authors are using are
concentrations and not doses. Please rearrange the paragraph
accordingly. I am referring to paragraph 2.3. Dose response analysis of
hit compounds.
We rearranged the paragraph as suggested
This manuscript is a resubmission of an earlier submission. The following is a list of the peer review reports and author responses from that submission.
Round 1
Reviewer 1 Report
Revision molecules-491578:
The authors are here reporting a manuscript entitled “Focus library of quinoxaline 1,4-di-N-oxide derivatives as cathepsin L inhibitors with activity against the liver fluke”. The work overall is well conducted although there are some major experiments that are missing.
Major concerns:
Table 3: please extend the HepG2 IC50 determination also to the not determined compounds.
IC50 values against parasites vitality should be determined.
Table 1 and related manuscript: It is very complex to define structure activity relationship by using information on a single concentration. As can be noted by the reported IC50 some of the % of inhibition do not necessarily match with higher or lower IC50. To this extend I would suggest to eliminate or at least significatively reduce the paragraph where SAR are reported.
Minor concerns:
Figure 4: It is not clear what X-axes means.
Each figure or table should be enriched with statistics data including standard deviation, information over the procedures used, and experimental procedure parameters (e.g. Table 1, add the concentration used for the screening).
Table 1; it is very hard to define compounds in table 1. Please if possible review this table. Add whole chemical structures in supporting information.
Page 7 line 171: to my opinion what the authors are using are concentrations and not doses. Please rearrange the paragraph accordingly.
Author Response
Thanks for your valuable comments. Please find our response next to your suggestions:
Table 3: please extend the HepG2 IC50 determination also to the not determined compounds.
We didn't performed the IC50 experiments with C7 and C19 because we don't have enough amount of those compounds to test. We are planning to re-synthetize some of the most promisin compounds to continue our experiments, but C7 didn't kill the NEJ so for now to re/synthetize that compound is not a priority. We will synthetize C19 but it takes time and we think that as a primary screening of the compounds citotoxicity we have data for several representative molecules. We erased the non determined compounds from Table 3.
IC50 values against parasites vitality should be determined.
It is very difficult and expensive to obtain metacercarie to perform the experiments with the NEJ, that's why we evaluated the compounds with the parasites at a fixed dose. There are very few metacercarie suppliers in the world and currently, our supplier in Uruguay, is not able to provide us more metacercarie.
Table 1 and related manuscript: It is very complex to define structure activity relationship by using information on a single concentration. As can be noted by the reported IC50 some of the % of inhibition do not necessarily match with higher or lower IC50. To this extend I would suggest to eliminate or at least significatively reduce the paragraph where SAR are reported.
We agree with the comments but also believe that some roughly observations can be made. Following your recommendation, we erased the second paragraph of the SAR discussion. We kept the first one because we consider that even if we used a single concentration, the observed tendencies are based on the activity of several compounds.
Minor concerns:
Figure 4: It is not clear what X-axes means.
We used the log of the concentration of the compounds to fit the data to a sigmoid curve to calculate the IC50.
Each figure or table should be enriched with statistics data including standard deviation, information over the procedures used, and experimental procedure parameters (e.g. Table 1, add the concentration used for the screening).
We added this information where it was missing
Table 1; it is very hard to define compounds in table 1. Please if possible review this table. Add whole chemical structures in supporting information.
We organized Table 1 showing the R1-R4 substituents to facilitate the comparison of the compounds structures, to better see which are the shared substituents. We added a supporting Figure with the whole chemical structures as suggested.
Page 7 line 171: to my opinion what the authors are using are concentrations and not doses. Please rearrange the paragraph accordingly.
The word dose is used several times at page 8 when discussing about IC50 values, but not in page 7, what are you refering to?
Reviewer 2 Report
In this manuscript Florencia Ferraro and coworkers identified and characterized quinoxaline 1,4-di-N-oxide derivatives as novel inhibitors of the two main cathepsins secreted by juvenile and adult liver flukes. Molecular docking and molecular dynamics simulations were also performed to predict the binding site and the binding interactions of the quinoxalines with cathepsins.
This paper describes biological results of interest. The research methodologies and methods for the study are appropriate. The paper is well organized but it still needs some minor modifications.
In figure 7 the authors should check the colours of the figures. They use different colours in the caption. The figures on the right should be changed to be clearer. Amino-acid labels should be moved to avoid overlapping with the drawings. Besides, the use of different colours for the ligand and for the amino-acid residues of the enzyme would improve the figure.
Author Response
Thanks for your helpfull comments.
We modified the figure according to your suggestions.

Round 2
Reviewer 1 Report
Revision molecules-491578:
Very little has been done to improve the quality of the manuscript.
Indeed, the authors report “It is very difficult and expensive to obtain metacercarie to perform the experiments with the NEJ, that's why we evaluated the compounds with the parasites at a fixed dose. There are very few metacercarie suppliers in the world and currently, our supplier in Uruguay, is not able to provide us more metacercarie.” This is a very impressive response. I believe that a trained scientist should be able to schedule and program the experiments and to acquire the necessary materials in order to complete that certain experiment. In absence of more in depth experiments over parasites for at least one optimal compound, the information collected so far in this manuscript are too preliminary and publication is not recommended.
Second reply: “We agree with the comments but also believe that some roughly observations can be made. Following your recommendation, we erased the second paragraph of the SAR discussion. We kept the first one because we consider that even if we used a single concentration, the observed tendencies are based on the activity of several compounds”. It is not possible to extrapolate such a kind of SAR comments as those reported in the manuscript by using a single concentration. This part of the manuscript must be removed or significantly reduced.
“We used the log of the concentration of the compounds to fit the data to a sigmoid curve to calculate the IC50”. So it has been used the nanoMolar log concentration? This is quite peculiar, usually log concentration are reported in Molar concentration.
Page 7 line 171: to my opinion what the authors are using are concentrations and not doses. Please rearrange the paragraph accordingly. I am referring to paragraph 2.3. Dose response analysis of hit compounds.